# Is There a Relationship Between Vitamin D Deficiency and Primary Monosymptomatic Enuresis Nocturna?

**DOI:** 10.3390/diagnostics15111345

**Published:** 2025-05-27

**Authors:** Abdulgani Gülyüz

**Affiliations:** Department of Pediatrics, School of Medicine, Turgut Ozal University, 06560 Malatya, Turkey; abdulganigulyuz@gmail.com

**Keywords:** primary monosymptomatic enuresis nocturna, vitamin D deficiency, child, associated factors, bedwetting

## Abstract

**Objective:** The aim of this study was to investigate the relationship between primary monosymptomatic enuresis nocturna (PMNE) and vitamin D deficiency in children. **Patients and Methods:** This retrospective case–control study included 307 PMNE patients aged 5–18 years and 254 age- and sex-matched healthy control subjects. Demographic data and biochemical parameters of the participants were obtained from hospital records. Serum 25(OH)D3 levels were measured using the chemiluminescence immunoassay method. The Mann–Whitney U test, Chi-square test, Pearson correlation and multivariate logistic regression analysis were used for statistical analyses. **Results:** Serum 25(OH)D3 levels were significantly lower in the PMNE group compared to the control group (*p* < 0.001). The rate of vitamin D deficiency was higher in the PMNE group. Vitamin D deficiency (OR: 3.164, 95% CI: 1.195–8.378, *p* = 0.02) and family history of enuresis (OR: 2.790, 95% CI: 1.01–5.8, *p* = 0.04) were found to be independent associated factors for PMNE. A significant negative correlation was found between serum vitamin D level and weekly bedwetting frequency (r = −0.377, *p* < 0.001). **Conclusions:** Serum 25(OH)D3 levels were significantly lower in the PMNE group (*p* < 0.001, Cohen’s d = 0.89). It is recommended that vitamin D levels should be routinely evaluated in children with PMNE and the potential benefits of vitamin D supplementation should be investigated in prospective studies.

## 1. Introduction

Primary monosymptomatic enuresis nocturna (PMNE) is a clinical picture characterised by urinary incontinence during sleep without daytime urinary symptoms and is seen in children who have completed the age of five years. This condition is one of the most common urinary incontinence problems of childhood and may negatively affect the quality of life of both the child and his/her family [1,2]. PMNE may be attributed to multifactorial causes including bladder dysfunction, increased urine production and disorders in the waking mechanism [3,4,5,6,7,8].

In recent years, studies focusing on the role of metabolic and neurological factors in the pathophysiology of PMNE have pointed out the importance of vitamin D [7,8]. The presence of vitamin D receptors in the central nervous system and bladder tissue suggests that this vitamin may have an indirect role in urinary control [9]. In addition, it has been reported that vitamin D deficiency is associated with sleep disorders and disruption of sleep patterns may constitute an associated factor for the development of PMNE [10]. In recent years, there have been few studies investigating the relationship between PMNE and vitamin D deficiency [11,12]. Siroosbakht et al. [13] found that vitamin D levels were lower in PMNE patients compared with healthy children. However, the number of patients in the study was small and confounding factors were not sufficiently discussed.

In this context, our study, one of the largest to date, was designed to test the hypothesised relationship between vitamin D deficiency and PMNE based on prior studies, using an existing database.

## 2. Patients and Method

This study was carried out through retrospective analysis of the data of the patients who applied to the Pediatric Outpatient Clinic of Malatya Training and Research Hospital between 1 June 2010 and 1 January 2015.

Inclusion criteria: 5–18 years of age and a diagnosis of primary PMNE.

Exclusion criteria: Secondary enuresis, neurological diseases, chronic constipation, urinary tract infections, anatomical urinary tract anomalies, obesity, malnutrition, diabetes, steroid use, sleep disorders, current use of vitamin D supplements and psychosocial disorders.

Age, gender, height, weight, history of premature birth, history of allergy, family history of enuresis, parental education level, frequency of nocturnal bedwetting (number of nights per week) and laboratory parameters (serum 25(OH)D3, parathormone, calcium, phosphorus, alkaline phosphatase, serum iron and magnesium) were collected retrospectively from hospital records.

A total of 349 children who presented to the outpatient clinic between 1 June 2010 and 1 January 2015 were evaluated with suspicion of MNE. In total, 42 patients were excluded from this study due to neurological disease, chronic constipation, history of vitamin supplementation, etc. According to the International Child Continence Society (ICCS) criteria, 307 patients were included in the case group (5–18 years of age and diagnosed with primary MNE). In the same period, 275 healthy children who were in follow-up were evaluated as control candidates. The control group consisted of healthy children who were admitted to the hospital for healthy child follow-up or with complaints other than enuresis. In our country, health expenditures are fully covered by the state, and no other financing source was used for the examinations and imaging performed. Twenty-one children were excluded from the control group due to malnutrition, obesity and vitamin D supplement use. As a result, 254 healthy children were included in the control group (Figure 1). The control group consisted of healthy children with no history of enuresis, confirmed through parental interview at the time of hospital admission.

**Laboratory analyses:** Serum 25(OH)D3 and parathormone levels were measured using the chemiluminescence immunoassay method using a Roche Cobas e601 device. Other biochemical parameters were analysed using spectrophotometric methods.

Classification of vitamin D levels: ≥20 ng/mL is considered normal, 12–20 ng/mL is considered insufficient and <12 ng/mL is considered deficient [14,15].

Statistical analyses:

In the sample size analysis performed with G Power, it was determined that a sample of at least 242 people was needed in each group when set as “two-way”, “effect size = 0.3”, “α = 0.05” and “1 − β = 0.95”. In this study, this number was exceeded in both groups to obtain stronger results. The sample size calculation was based on detecting a significant difference in group means of serum 25(OH)D3 levels.

The IBM SPSS Statistics 23.0 programme was used. The distribution of continuous variables was evaluated using the D’Agostino–Pearson test. The Mann–Whitney U test was used for non-normally distributed data. Categorical variables were analysed using the Chi-square test. The relationship between serum vitamin D level and weekly frequency of bedwetting was evaluated using Pearson correlation analysis. Multivariate logistic regression analysis was used to determine the independent factors increasing the risk of PMNE. Initially, univariate logistic regression analysis (enter method) was performed for all candidate explanatory variables. Variables with a *p*-value < 0.05 were included in the multivariate logistic regression using the backward LR method. All included and excluded variables, as well as non-significant results, are presented and discussed. *p* < 0.05 was considered significant. Multivariate logistic regression analysis was performed using weekly enuresis frequency as the dependent variable. The mean number of enuretic events per night was assessed separately for descriptive purposes. The variables included in this model were as follows: serum vitamin D level, serum iron, serum magnesium level, family history of enuresis, parental education level, history of allergy, gender, BMI and age group. In this way, the effect of vitamin D level on PMNE was tested independently of other confounding variables. Pre-specified hypothesis testing (e.g., comparison of vitamin D levels between groups) and exploratory analyses (e.g., logistic regression) were clearly distinguished and reported accordingly.

Demographic, clinical and laboratory data of all participants included in this study were complete. Individuals with missing data were excluded at the beginning of this study. Therefore, there were no missing data in the analyses.

Although a broader panel of laboratory markers was measured, only the parameters deemed clinically and statistically relevant to the scope of this study are included in the main tables. Other data are available upon request.

Ethical approval: The study was conducted in accordance with the Declaration of Helsinki and approved by the Ethics Committee of Malatya Turgut Özal University (Decision No: E-30785963-020-192067; Date: 22 February 2024). Due to its retrospective nature, patient consent was waived by the committee.

## 3. Findings

In this study, 307 PMNE patients and 254 healthy children aged 5–18 years were included. Demographic characteristics, laboratory findings and clinical parameters were compared between the PMNE group and the control group.

### 3.1. Demographic Characteristics

No statistically significant difference was found between the PMNE group and the control group in terms of age, gender, height, weight and history of premature birth (*p* > 0.05). The mean age of the PMNE group was 8 (5–17) years, while the mean age of the control group was 7 (5–15) years. The male/female ratio was similar in both groups (PMNE group: 174/133; control group: 142/112; *p* = 0.913). In addition, the rate of history of enuresis in first- and second-degree relatives was significantly higher in the PMNE group (35.2%) compared to the control group (3.9%) (*p* < 0.001) (Table 1).

### 3.2. Laboratory Findings

The serum vitamin D level (median: 16 ng/mL) was significantly lower in the PMNE group compared to the control group (median: 25.5 ng/mL) (*p* < 0.001). In addition, the parathormone level (median: 29 pg/mL) was higher in the PMNE group compared to the control group (median: 27 pg/mL) (*p* < 0.001). The high PTH level found in the PMNE group may be a biochemical reflection of vitamin D deficiency in these children. In addition to *p*-values, effect sizes (Cohen’s d) were calculated to assess the magnitude and clinical relevance of the differences between groups. No statistically significant difference was found between the two groups in terms of calcium, phosphorus, alkaline phosphatase, serum iron and magnesium levels (*p* > 0.05) (Table 2).

### 3.3. Distribution of Vitamin D Levels

Vitamin D levels were categorised as adequate (20–100 ng/mL), insufficient (12–20 ng/mL) or deficient (<12 ng/mL) per the Global Rickets Guidelines. In the PMNE group, 72% of children had insufficient or deficient levels, compared to ~29% in the control group (*p* < 0.001) (Table 3, Figure 2).

### 3.4. The Relationship Between Incontinence Frequency and Vitamin D Levels

A significant negative correlation was found between the number of weekly night wetting and vitamin D level (r = −0.377, *p* < 0.001) (Figure 3). This statistically significant finding indicates that bedwetting frequency increases as vitamin D level decreases. However, no significant correlation was found between the number of bedwetting incidents per night and vitamin D level (r = −0.126, *p* = 0.184) (Table 4).

### 3.5. Multivariate Logistic Regression Analysis

In the multivariate logistic regression analysis, independent associated factors affecting the development of PMNE were evaluated. All variables were initially evaluated by univariate logistic regression analysis. Variables with *p* < 0.05 were then included in the multivariate logistic regression model. The model included serum vitamin D level, age, gender, BMI, serum iron, parental education level and family history of enuresis. The regression model was constructed based on weekly enuresis frequency. No separate regression model was created for mean nightly frequency, which was only evaluated descriptively.

Vitamin D deficiency was found to be a significant associated factor for the development of PMNE (OR: 3.164; 95% CI: 1.195–8.378; *p* = 0.020). This finding suggests that individuals with low vitamin D levels have an approximately 3-fold increased risk of developing PMNE.

A family history of enuresis was also found to be a significant associated factor (OR: 2.790; 95% CI: 1.01–5.8; *p* = 0.042). This result supports that genetic predisposition plays an important role in the pathogenesis of PMNE.

Other variables included in the model, age (*p* = 0.412), gender (*p* = 0.598), BMI (*p* = 0.408), serum iron level (*p* = 0.327) and parental education level (*p* = 0.643), were not found statistically significant (Table 5).

## 4. Discussion

In this study, a significant relationship was found between vitamin D deficiency and PMNE in children. Serum vitamin D levels of children in the PMNE group were found to be significantly lower compared to the control group (*p* < 0.001). Although statistically significant differences were observed in vitamin D levels between groups, effect size measures were also considered to evaluate the clinical relevance of this finding. In addition, vitamin D deficiency was found to be an independent associated factor, increasing the risk of MNE 3.164-fold (OR: 3.164, 95% CI: 1.195–8.378, *p* = 0.020). These findings suggest that vitamin D deficiency may play an important role in the pathophysiology of MNE. Although a significant negative correlation was found (r = −0.377), the corresponding r² value (~0.14) indicates that only about 14% of the variance in PMNE can be explained by serum vitamin D levels, suggesting a modest effect size.

The relationship between vitamin D deficiency and enuresis nocturna has been a subject of increasing interest in recent years [11,12,13]. Vitamin D is known to have extensive effects on the central nervous system and peripheral tissues [16]. In particular, it has been shown to affect neurological functions by regulating the synthesis of neurotransmitters such as dopamine and serotonin. Vitamin D receptors are also expressed in bladder smooth muscle and urothelium. It has been suggested that vitamin D reduces contractions by suppressing sensory signals during the filling phase of the bladder, thereby reducing the risk of urinary incontinence [8,13,17,18,19,20].

In this study, the higher frequency of enuresis nocturna in children with vitamin D deficiency supports this mechanism. In addition, it is thought that vitamin D deficiency may increase urine production by affecting the expression of renal factors such as endothelin-1 [21,22]. In addition, the immunomodulatory effects of vitamin D may also play a role in the pathophysiology of PMNE [10,23]. Vitamin D may lead to sleep disorders by decreasing the release of inflammatory mediators such as tumour necrosis factor-alpha (TNF-α) and prostaglandin D2 [20,21,22]. This may be associated with poor sleep quality in children with NE. In this study, the finding that vitamin D deficiency was associated with enuresis nocturna suggests the potential effects of vitamin D on bladder control and sleep patterns.

Limited sunlight exposure, influenced by reduced outdoor activity or underlying conditions, may independently lower vitamin D levels. This potential confounder should be considered when interpreting the association with PMNE.

The rate of enuresis history in first- and second-degree relatives in the PMNE group was 35.2%, whereas this rate was 3.9% in the control group (*p* < 0.001). These findings show that family history is an important associated factor for PMNE. In the literature, the rate of positive family history has been reported between 27.7% and 64.8% [24,25]. Our study found that family history increased the risk of PMNE by 2.79 times, which is consistent with the literature. In our study, the male to female ratio was found to be 1.3, which is compatible with the male predominant distribution reported in the literature [1,24,25,26]. However, our study has a retrospective design and cannot be considered as a prevalence study.

Li et al. [11] reported vitamin D deficiency in one-third of enuresis nocturna cases in a study conducted in Chinese children. Hoda et al. [12] showed that 80% of patients with PMNE had vitamin D insufficiency and deficiency. In our study, serum vitamin D levels of children in the PMNE group were found to be significantly lower compared to the control group (*p* < 0.001). Additionally, vitamin D deficiency was found to be an independent associated factor that increases the risk of PMNE by 3.164 times (OR: 3.164, 95% CI: 1.195–8.378, *p* = 0.020). According to our data, deficiency was detected in 51.8% of children and insufficiency in 20.2%. These results are compatible with the literature. Similarly, some studies have shown a relationship between vitamin D deficiency and sleep disorders [19,27,28]. These studies suggest that vitamin D may affect sleep quality through receptors located in brain regions that control sleep patterns such as the hypothalamus. However, sleep quality parameters could not be analysed in our study because polysomnographic evaluation was not performed.

A significant negative correlation was found between weekly enuresis frequency and vitamin D levels (r = −0.377, *p* < 0.001), suggesting increased incontinence with lower vitamin D. This relationship was not observed with nightly frequency, possibly due to reporting differences or variability in symptoms. Siroosbakht S et al. [13] also reported a similar strong correlation. Hoda et al. [12] reported a significant correlation between the number of urinary incontinence episodes per night and vitamin D levels. However, no statistically significant correlation was found between these parameters in our study.

Environmental factors such as caffeine consumption may also affect the development of enuresis. In a recently published randomised controlled study, limited caffeine consumption (no history of caffeine consumption was obtained in our study) was shown to be effective in the treatment of PMNE [29,30]. In this context, it is important to evaluate both environmental and biochemical factors together.

### 4.1. Limitations of the Study

One of the strengths of our study is that it is the largest sample study on this subject so far. However, there are also some limitations. Firstly, it is not possible to establish a causal relationship due to the retrospective design of this study. The patient and control groups could not be compared in terms of factors such as socioeconomic status, dietary habits, caffeine consumption and duration of exposure to sunlight. It should be considered that these factors may affect vitamin D levels. This study included 307 patients and 254 control subjects. Although the sample size is sufficient, further studies in larger populations will increase the generalisability of the findings. Propensity score matching was not applied due to the retrospective nature of the dataset, which should be acknowledged as a methodological limitation.

Secondly, the lack of polysomnography limited the evaluation of sleep-related factors. The effect of vitamin D supplementation on PMNE was also not assessed. Although the regression model included key variables such as age, gender, vitamin D, serum iron and family history, other potentially relevant factors (e.g., sleep disorders, psychosocial stress, diet, fluid intake and physical activity) could not be evaluated due to the retrospective design, representing a limitation.

Finally, the fact that this study was conducted in a single centre in eastern Turkey limits the generalisability of the findings to populations in other geographical regions.

### 4.2. Suggestions for Future Studies

In the future, randomised controlled trials investigating the effect of vitamin D supplementation on PMNE are recommended. Mechanistic studies that further investigate the effects of vitamin D on bladder function and sleep patterns are also important. Such studies may help to better understand the mechanisms underlying the relationship between vitamin D deficiency and PMNE.

## 5. Conclusions

In our study, vitamin D deficiency was found to be an independent factor associated with PMNE and was negatively correlated with weekly wet night frequency. This study suggests that vitamin D deficiency may be associated with PMNE. However, further studies are needed to understand the causality and underlying mechanisms of this relationship.

## Figures and Tables

**Figure 1 diagnostics-15-01345-f001:**
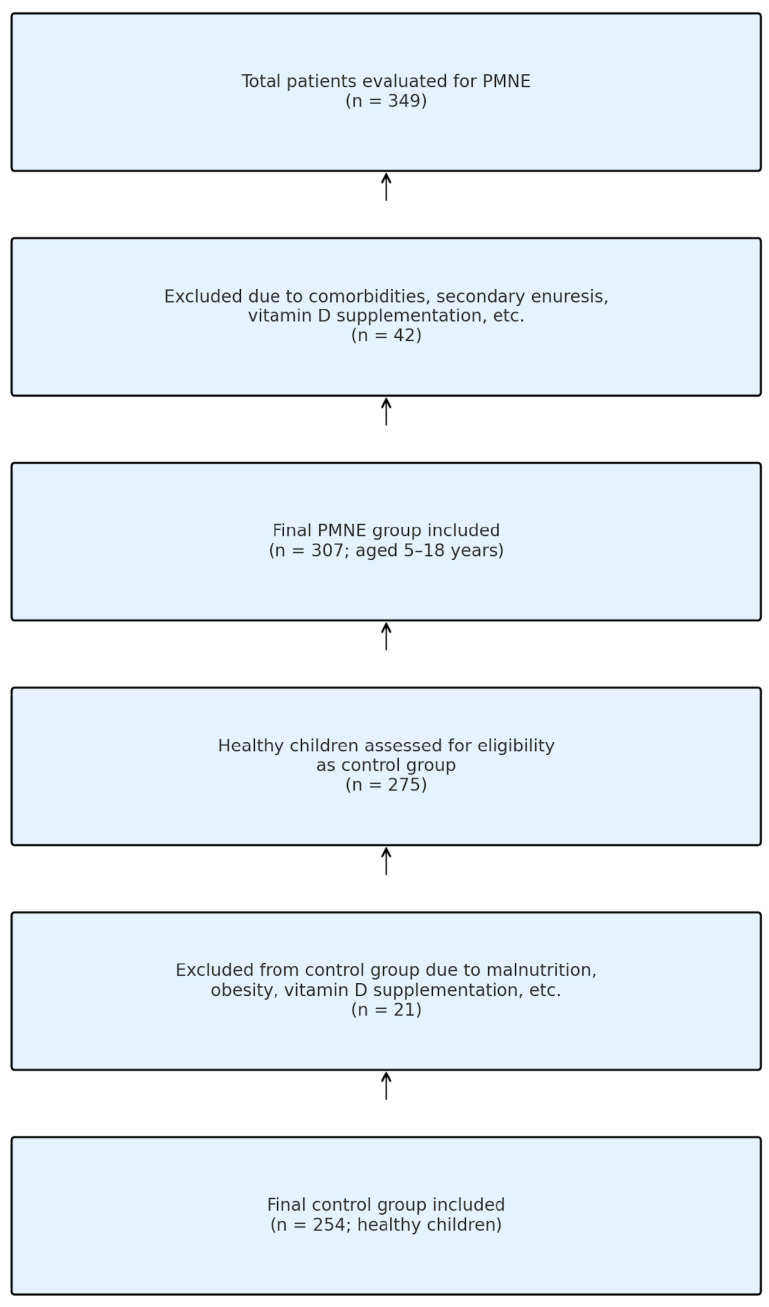
Flowchart of participant selection.

**Figure 2 diagnostics-15-01345-f002:**
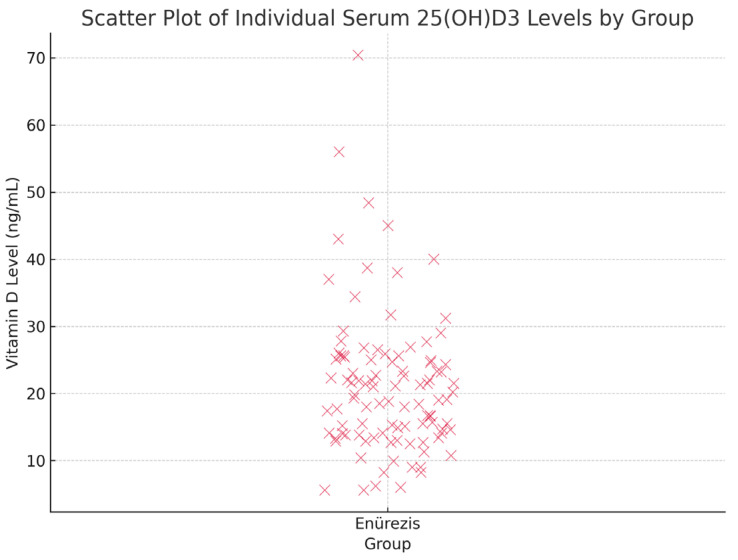
Scatter plot showing individual serum 25(OH)D3 levels in children with PMNE.

**Figure 3 diagnostics-15-01345-f003:**
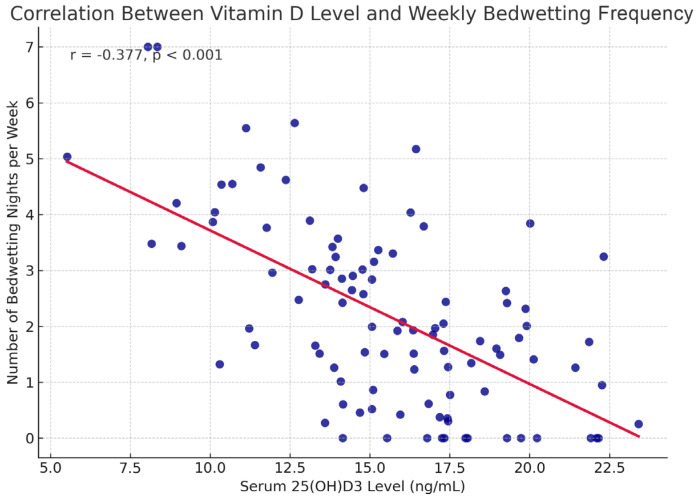
25(OH)D3 levels and the number of bedwetting nights per week in children with PMNE.

**Table 1 diagnostics-15-01345-t001:** Comparison of demographic characteristics of the groups.

Feature	Patient Group (*n* = 307)	Control Group (*n* = 254)	*p*
Gender (M/F) a	174/133	142/112	0.913
Age (years) b	8 (5–17)	7 (5–15)	0.176
Height (cm) b	125 (102–172)	122.5 (97–161)	0.215
Weight (kg) b	29.7 (14.7–73.4)	32.4 (16.1–65.6)	0.327
BMI	13.0 (12.8–21.1)	13.2 (12.8–22.0)	0.128
History of premature birth	31 (%10.1)	24 (%9.4)	0.195
History of allergy	39 (12.7)	27 (%10.6)	0.531
Mother’s education level (high school, %)	%22.7	%23.2	0.825
Father’s education level (high school %)	%26.1	%26.8	0.792
Family history of PMNE	108 (%35.2)	10 (%3.9)	<0.001 *
Weekly frequency of bedwetting	3.3 (1–7)	-	-
Number of bedwetting per night	1.2 (1–3)	-	-

a: Chi-square test (*n*, %), b: Mann–Whitney U test (median [min-max]), * *p* < 0.05 is significant. Data are presented as mean ± standard deviation for normally distributed variables, and as median (minimum–maximum) for non-normally distributed variables.

**Table 2 diagnostics-15-01345-t002:** Comparison of laboratory data.

Parameter	Patient Group (*n* = 307)	Control Group (*n* = 254)	*p*
25(OH)D3 (ng/mL)	16 (4–37)	25.5 (15–48)	<0.001 *
Parathormone (pg/mL)	29 (7–77)	27 (12–65)	<0.001 *
Calcium (mg/dL)	9.6 (8.7–10.6)	9.8 (8.9–11.4)	0.076
Phosphorus (mg/dL)	4.8 (3.7–6.8)	4.7 (3.5–8.5)	0.066
Alkaline Phosphatase (IU/L)	229 (120–485)	210 (124–383)	0.189
Magnesium (mg/dL)	1.9 (1.2–2.3)	1.8 (1.1–2.2)	0.111
Serum Iron (µg/dL)	55 (55.1 ± 14.8)	57 (56.8 ± 15.3)	0.546

Mann–Whitney U test, * *p* < 0.05 is significant.

**Table 3 diagnostics-15-01345-t003:** Distribution according to vitamin D levels.

Vitamin D Level	Patient Group (*n* = 307)	Control Group (*n* = 254)	*p*
Adequate (20–100 ng/mL)	86 (%28)	181 (%71.3)	<0.001
Insufficient (12–20 ng/mL)	159 (%51.8)	63 (%24.8)	<0.001
Deficient (<12 ng/mL)	62 (%20.2)	10 (%3.9)	<0.001

Chi-square test. *p* < 0.001.

**Table 4 diagnostics-15-01345-t004:** Correlation between frequency of urinary incontinence and vitamin D level.

Variables	Correlation Coefficient (r)	*p*
Number of enuresis per night— vitamin D level	−0.126	0.184
Number of bedwetting nights per week— vitamin D level	−0.377 **	<0.001 *

* *p* < 0.05 was accepted as significant. ** Significant negative correlation.

**Table 5 diagnostics-15-01345-t005:** Independent associated factors for PMNE (logistic regression result).

Variable	OR (95% CI)	* *p*-Value
Vitamin D deficiency	3.164 (1.195–8.378)	0.020
Family history of enuresis	2.790 (1.01–5.8)	0.042
Age (years)	1.02 (0.96–1.08)	0.410
Gender (M/F)	1.12 (0.74–1.69)	0.643
BMI	0.97 (0.89–1.06)	0.408
Serum iron (µg/dL)	0.99 (0.97–1.02)	0.327
Parental education level (high school and above)	0.91 (0.61–1.36)	0.643

* *p* < 0.05 was considered statistically significant. BMI, body mass index.

## Data Availability

The original contributions presented in the study are included in the article, further inquiries can be directed to the corresponding author.

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
