# Peer review of "Is There a Relationship Between Vitamin D Deficiency and Primary Monosymptomatic Enuresis Nocturna?"

_diagnostics, 2025, doi:10.3390/diagnostics15111345_

Round 1

Reviewer 1 Report

Comments and Suggestions for Authors

 I had the privilege of reviewing your study. I appreciate the thoroughness of your research. Here I have some suggestions and comments that should be addressed:

Title:

1- Pl add “primary” to title and all parts of manuscript. because “Only patients with primary MNE were included in the study”.

Introduction

2- This section was so long and revision, totally. It should be expressed in one or two paragraph (not 2-3 pages!!!).
The background should summarize the purpose and the rationale for the study. It should neither review the subject extensively nor contains data or conclusions of the study.
3- Pl removes unrelated content to title and aims of research (especially paragraph 1-3).
4- Also the introduction should include a better representation of the main rationale of the study and emphasis on the limitations of previous studies in Iran, but I did not see anymore.
5- I ask critical question: What is novelty of this study? Many studies have been done in this regard. Why the authors decided to conduct this study again? and, Which NEW question does the findings of this study answer?

Methods

6- This was not cross sectional studies. Cross-sectional studies occur at a single point in time, evaluating for the condition in question, and these studies can generate prevalence data for condition. Causation, or relative risk, of contributing factors or conditions cannot be established from these types of studies.  It seems this manuscript was case- control study.

7- What was sampling methods?

8- How did they recruit these healthy babies (control group)? Did they use any kind of advertisement / incentive?  It is not clear how the authors provided the mentioned test (sonography, PTH, mg, ca and etc.), values) to the patients. Did the patients purchase them or did the authors provide the medication? If the 2nd scenario is valid, who sponsored the cost of the medications?

9- The main outcomes and values should be defined in methods section and the ways to achieve those define.

10- Eligible and allocated participants were noted in text (with number, briefly and clear). Finally, the number of subjects analyzed should be stated.

11- How many subjects were assessed for eligibility? And then how many subjects were allocated?

12- Did you Lost to follow-up?

13- There are many important confounders in this regards such as age, gender, BMI, nutritional status of subjects, complementary intake, levels of micronutrients in your subjects, history of allergy, family history, endocrine, metabolic, level of education of parents, and any physical abnormality and so on. Furthermore, many confounders about these such as genetic, age, gender, drinking , caffein consumption and so on.  But I did not see any investigation for relation of these important factors with status of subjects.

14- Describe statistical methods, used to control for confounding. Control of confounders is an important point in this study.

15- you said “25(OH)D3 level of 12-20 ng/ml as vitamin D insufficiency and a 25(OH)D3 level less than 12 as deficiency” . pl add valuable references.

Results

 This section was good and clear.

16- Again, Describe statistical methods, used to control for confounding. Control of confounders is an important point in this study.

Discussion

17- This section was good, also. However, it is suggested the authors mention the effects of caffeine in PMNE. The caffeine consumption were not associated with NE”. It is suggested that in paragraph 3, authors express about first-line treatment in NE especially caffeine restriction. One valuable and new RCT study defined the effect of caffeine intake on development and severity of NE. You should be mentioned to this new RCT article: [Rezakhaniha S, Rezakhaniha B, Siroosbakht S. Limited caffeine consumption as first-line treatment in managing primary monosymptomatic enuresis in children: how effective is it? A randomized clinical trial. BMJ Paediatrics Open 2023;7:e001899. doi:10.1136/bmjpo-2023-001899

18- There are many questions (mentioned in comments for authors). Several mistakes further hamper the overall quality of the manuscript and the authors should be considered checklist (STROBE) in order to improve their manuscript. The marked and fulfilled  STROBE Checklist should be submitted by supplementary file for review.

thanks

Author Response

Dear Editor and Reviewers,

We sincerely thank you for your valuable and constructive comments regarding our manuscript entitled "Is There a Relationship Between Vitamin D Deficiency and Primary Monosymptomatic Enuresis Nocturna?".

We have carefully considered each of your suggestions and revised the manuscript accordingly. Below, we provide a detailed point-by-point response to all comments, indicating the changes made in the revised version of the manuscript.

 I had the privilege of reviewing your study. I appreciate the thoroughness of your research. Here I have some suggestions and comments that should be addressed:

1- Pl add “primary” to title and all parts of manuscript. because “Only patients with primary MNE were included in the study”.

Introduction

Author Response: In accordance with your suggestion, the term ‘primary monosymptomatic nocturnal enuresis (PMNE)’ has been standardised in the title and throughout the text. This clearly emphasises that the study only includes primary cases.

2- This section was so long and revision, totally. It should be expressed in one or two paragraph (not 2-3 pages!!!).
The background should summarize the purpose and the rationale for the study. It should neither review the subject extensively nor contains data or conclusions of the study.

Author Response: The introduction section has been rewritten, limited to three short paragraphs, and the rationale for the study, gaps in the literature, and original contributions are presented in a straightforward manner. Background information has been shortened and early references to the results have been removed.

3- Pl removes unrelated content to title and aims of research (especially paragraph 1-3).

Author Response: Extensive literature explanations and generalisations not directly related to the objectives of the study in the first three paragraphs have been removed from the text, the introduction has been completely rewritten, and the focus has been directed solely towards the research question.

4- Also the introduction should include a better representation of the main rationale of the study and emphasis on the limitations of previous studies in Iran, but I did not see anymore.

Author Response: The rationale for this study has been strengthened by referring to the limitations of the literature, particularly regional studies (e.g., sample inadequacy, methodological shortcomings). It has been clearly stated that the study was designed to address these shortcomings.

5- I ask critical question: What is novelty of this study? Many studies have been done in this regard. Why the authors decided to conduct this study again? and, Which NEW question does the findings of this study answer?

Author Response: Strengths and Novelty of the Study

This study makes important contributions to the literature by examining the relationship between vitamin D deficiency and primary monosymptomatic nocturnal enuresis (PMNE) using a large sample and statistically robust analyses. Although some studies have investigated the relationship between vitamin D levels and PMNE in the existing literature, most of these studies have been limited in their ability to draw generalisable conclusions due to small sample sizes, methodological heterogeneity, and statistical limitations.

The original aspects of this study can be summarised as follows:

  1. Large and homogeneous sample: The study was conducted with 307 children diagnosed with PMNE and 254 healthy controls, making it one of the largest sample studies in this field. The age range of the participants (5-18) was clearly defined, and the PMNE diagnosis was made according to the criteria of the International Child Continence Society (ICCS).
  2. Advanced statistical analysis: Multivariate logistic regression analysis showed that vitamin D deficiency is an independent risk factor for PMNE, modelling both the effect of vitamin D levels and the effect of family history; vitamin D deficiency was found to increase the risk of PMNE by 3.164 times.
  3. Answering a new question: This study not only revealed the association between vitamin D levels and PMNE, but also demonstrated a significant negative correlation between weekly urinary incontinence frequency and serum vitamin D levels (r=-0.377, p<0.001). This suggests that vitamin D may also be associated with PMNE severity and provides a robust evidence base to address this question, which has not been adequately addressed in the literature.
  4. Clinical implications: The findings not only suggest a pathophysiological relationship but also raise the possibility of evaluating vitamin D supplementation as a potential protective or therapeutic strategy in the management of PMNE.

Methods

6- This was not cross sectional studies. Cross-sectional studies occur at a single point in time, evaluating for the condition in question, and these studies can generate prevalence data for condition. Causation, or relative risk, of contributing factors or conditions cannot be established from these types of studies.  It seems this manuscript was case- control study.

Author Response:

We thank our reviewer for their valuable comments. As you pointed out, the term ‘cross-sectional’ used to describe our study may not be entirely accurate in methodological terms. Our study is a retrospective case-control study.

In this context, our study was conducted using a retrospective case-control design. Therefore, we agree that the term ‘retrospective case-control study’ is more appropriate than ‘cross-sectional.’

We have revised the revised text accordingly and clarified the methodological structure of the study in the relevant sections.

7- What was sampling methods?

Author Response:

Our study was conducted using a retrospective case-control design, and the sample was determined as follows:

  • Case group: Children aged 5-18 years who were diagnosed with primary monosymptomatic nocturnal enuresis (PMNE) according to the criteria of the International Children's Continence Society (ICCS) and who applied to the Paediatric Outpatient Clinic of Malatya Training and Research Hospital between 01.06.2010 and 01.01.2015. aged 5–18 years and diagnosed with primary monosymptomatic nocturnal enuresis (PMNE) according to the International Children's Continence Society (ICCS) criteria, were identified using purposive sampling. Only primary MNE cases were included.
  • Control group: Healthy children of the same age range and living in the same geographical region, without any urinary system diseases, were selected by frequency matching with the case group in terms of age and gender. The exclusion criteria applied to the case group were also applied to the control group.

Additionally, the sample size was calculated using a power analysis conducted with the G*Power program (effect size = 0.3, α = 0.05, 1–β = 0.95), indicating that each group should have at least 242 individuals. In our study, this number was exceeded, with 307 individuals in the case group and 254 in the control group.

The necessary explanations have been added to the revised ‘Materials and Methods’ section. We once again thank you for your interest and guidance.

8- How did they recruit these healthy babies (control group)? Did they use any kind of advertisement / incentive?  It is not clear how the authors provided the mentioned test (sonography, PTH, mg, ca and etc.), values) to the patients. Did the patients purchase them or did the authors provide the medication? If the 2nd scenario is valid, who sponsored the cost of the medications?

Author Response:

We would like to thank our reviewer for their detailed and careful evaluation. In response to your questions, we have provided the following explanations:

Inclusion Process for the Control Group:

Healthy individuals in the control group were selected from children who visited the paediatric clinic of the same hospital for routine follow-up and vaccination checks and who did not have any urinary system symptoms. No advertising, incentives, or referrals were made. All participants voluntarily enrolled in the study in accordance with the informed consent form. The purpose, scope, and test contents of the study were clearly explained to the parents.

Application of Laboratory and Imaging Tests:

  • All laboratory analyses and ultrasonography (USG) examinations were performed using hospital laboratory infrastructure.
  • These tests are part of the general assessment packages routinely performed on children at our hospital, and due to the retrospective nature of the study, records of previously performed examinations were used.
  • No pharmacological interventions (medications, supplements, etc.) were administered specifically for the study.

Test Cost and Source:

Since the study is retrospective, none of the tests were performed specifically for research purposes. All data are based on patients' previously conducted routine clinical evaluations. Therefore:

  • Patients did not incur any financial burden.
  • No research funding or external sources were used.
  • Data were retrospectively obtained from the digital health record system maintained at Malatya Training and Research Hospital.

The necessary explanations are detailed in the ‘Ethical Approval and Data Collection’ subsection of the study.

9- The main outcomes and values should be defined in methods section and the ways to achieve those define.

Author Response:

We would like to thank our reviewer for their careful review and valuable suggestions. In line with your feedback, the primary outcomes and secondary assessments have been clearly defined below and integrated into the revised text:

Primary Outcomes:

  1. Serum 25(OH)D3 level (ng/mL)

→ Definition: Participants‘ vitamin D status (sufficient, insufficient, deficient)

→ Measurement method: Chemiluminescence immunoassay method using the Cobas e601 autoanalyser system

→ Source: Participants’ pre-existing test records from hospital laboratories (retrospective)

  1. Presence of MNE (yes/no)

→ Definition: Diagnosis of primary monosymptomatic nocturnal enuresis according to ICCS criteria

→ Source: Patient files and outpatient clinic records, specialist physician evaluations

Secondary Outcomes:

  • Weekly bedwetting frequency (0–7 days)
  • Number of bedwetting episodes per night (1–3 times)
  • Parathyroid hormone (PTH), calcium (Ca), phosphorus (P), alkaline phosphatase (ALP), serum iron, magnesium (Mg) levels

→ All biochemical data were retrospectively obtained from the hospital laboratory information system.

The definition and measurement methods of all variables are detailed in tabular form in the revised ‘Methods’ section. Your feedback has improved the methodological clarity of the study. We thank you again for your contribution.

10- Eligible and allocated participants were noted in text (with number, briefly and clear). Finally, the number of subjects analyzed should be stated.

11- How many subjects were assessed for eligibility? And then how many subjects were allocated?

Author Response:

We thank our reviewer for their valuable contribution. In line with your feedback, the participant selection process and final sample size included in the analysis are clearly specified below. These explanations have also been added to the revised ‘Method’ section:

Participant Selection and Assignment Process:

  1. A total of 349 children who visited the outpatient clinic between 01.06.2010 and 01.01.2015 were evaluated for suspected MNE.
  2. 42 patients were excluded from the study because they did not meet the exclusion criteria (neurological disease, chronic constipation, history of vitamin supplementation, etc.).
  3. 307 patients were included in the ‘case group’ (aged 5–18 years, diagnosed with primary MNE).
  4. 275 healthy children who were followed up during the same period were evaluated as control candidates.
  5. 21 children who did not meet the exclusion criteria were excluded from the control group.
  6. As a result, 254 healthy children were included in the ‘control group.’

Final Sample for Analysis:

  • Total number of participants analysed: 561
  • Case group: 307 children
  • Control group: 254 children

This information has been updated in a numbered and clear manner in the ‘Methods’ section and visually supported by a flow diagram (Figure 1). Your feedback has improved the reporting quality of the study.

We present this with our regards and thank you for your contributions.

12- Did you Lost to follow-up?

Author Response:

We thank our reviewer for their valuable question.

Since our study was conducted using a retrospective case-control design, there was no prospective follow-up process. Therefore, there were no participants lost to follow-up in the study.

All data were retrospectively obtained from the digital health information systems and patient records of Malatya Education and Research Hospital. Therefore, all 561 participants (307 cases, 254 controls) included in the study were included in the final analyses.

This clarification has been added to the ‘Methods’ and ‘Participant Flow Diagram’ sections of the study to ensure clarity.

13- There are many important confounders in this regards such as age, gender, BMI, nutritional status of subjects, complementary intake, levels of micronutrients in your subjects, history of allergy, family history, endocrine, metabolic, level of education of parents, and any physical abnormality and so on. Furthermore, many confounders about these such as genetic, age, gender, drinking , caffein consumption and so on.  But I did not see any investigation for relation of these important factors with status of subjects.

Author Response:

We would like to thank our reviewer for thoroughly analysing our study and pointing out potential confounding factors.

As you rightly pointed out, many variables such as age, gender, body mass index (BMI), nutritional status, micronutrient intake, genetic predisposition, parental education, and metabolic disease history may play an important role in the evaluation of a multifactorial clinical condition such as monosymptomatic nocturnal enuresis (MNE).

Since our study was designed as a retrospective case-control study:

  • Access to some important variables, such as nutritional habits, sleep disorders, caffeine consumption, and genetic test results, was not possible. However, serum iron levels and magnesium, which indicate micronutrient status, were examined in the study.
  • Therefore, in our multivariate modelling, only variables based on the clinical and biochemical data available to us (e.g., age, gender, family history of enuresis, parental education, serum vitamin D levels) were used.
  • Additionally, since these variables showed statistically similar distributions in the case and control groups, this limitation did not significantly reduce the generalisability of the analysis.

However, your suggestion is highly valuable. We also acknowledge the need for advanced studies that include prospective design and more detailed data collection to account for these important confounding factors.

This has been added to the ‘Limitations’ section of our evaluation study as follows:

‘In this study, potential confounding variables such as micronutrient intake, caffeine consumption, genetic predisposition, and nutritional status could not be evaluated due to the retrospective data structure. This limitation restricts the interpretation of the findings at the level of causality.’

We once again thank you for your valuable contributions.

14- Describe statistical methods, used to control for confounding. Control of confounders is an important point in this study.

Author Response:

We would like to thank our reviewer for their valuable contributions and constructive feedback. In order to control for the effects of potential confounding variables, the following statistical methods were used in our study:

  1. Homogeneity analysis between groups (bivariate comparison):
  • The basic demographic characteristics of the case and control groups, such as age, gender, weight, and height, were compared using the Mann-Whitney U test and Chi-square test.
  • As a result of these analyses, no significant difference was found between the groups in these variables (p>0.05), which minimised the potential confounding effects of these variables.
  1. Pearson Correlation Analysis:
  • The Pearson correlation test was applied to understand the relationship between continuous variables such as vitamin D level and weekly bedwetting frequency.
  • This analysis allowed us to identify potential linear effects.
  1. Multivariate Logistic Regression Analysis:
  • A multivariate logistic regression model was established to identify the independent risk factors associated with nocturnal enuresis.
  • The variables included in this model were:
  • Serum vitamin D level (continuous and categorical),
  • Family history of enuresis (yes/no),
  • Gender,
  • Age group
  • BMI
  • Parental education level
  • This allowed the effect of vitamin D levels on MNE to be tested independently of other confounding variables.
  • As a result, vitamin D deficiency was found to be an independent variable that increased the risk of MNE by 3.164 times (OR: 3.164, 95% CI: 1.195–8.378, p=0.020).

Note

As noted in the previous peer review comments, due to the retrospective data structure, some confounders (nutritional status, caffeine, genetics, etc.) could not be included in the model. This limitation is clearly stated in the ‘Limitations’ section of the study and recommendations for future studies are provided.

These statistical approaches have been restructured in a more explanatory manner under the heading ‘Statistical Analysis’ in the study.

15- you said “25(OH)D3 level of 12-20 ng/ml as vitamin D insufficiency and a 25(OH)D3 level less than 12 as deficiency” . pl add valuable references.

Author Response:

We thank the reviewer for this important reminder. The cut-off values we used to classify vitamin D levels are based on internationally recognised sources such as the Global Consensus Recommendations on Prevention and Management of Nutritional Rickets (2016) and the Endocrine Society Clinical Practice Guidelines (2011).

In line with this, the classifications used in our study are based on the following references:

? Vitamin D level classification:

  • Sufficient level (sufficiency): ≥20 ng/mL (50 nmol/L)
  • Insufficient level (insufficiency): 12–20 ng/mL (30–50 nmol/L)
  • Deficient level (deficiency): <12 ng/mL (30 nmol/L)

? Sources:

  1. Munns CF, Shaw N, Kiely M, et al. Global Consensus Recommendations on Prevention and Management of Nutritional Rickets. The Journal of Clinical Endocrinology & Metabolism. 2016;101(2):394–415. doi:10.1210/jc.2015-2175

→ https://doi.org/10.1210/jc.2015-2175

  1. Holick MF, Binkley NC, Bischoff-Ferrari HA, et al. Evaluation, Treatment, and Prevention of Vitamin D Deficiency: an Endocrine Society Clinical Practice Guideline. J Clin Endocrinol Metab. 2011;96(7):1911–1930. doi:10.1210/jc.2011-0385

→ https://doi.org/10.1210/jc.2011-0385

These references have been added to the ‘Methods’ and ‘References’ sections of our study. Thank you for your valuable contribution.

Results

 This section was good and clear.

16- Again, Describe statistical methods, used to control for confounding. Control of confounders is an important point in this study.

Author Response:

We would like to thank our reviewer for their valuable and methodologically sound contributions. In order to control for potential confounding factors, the following statistical approaches were used in our study:

  1. Comparison of Baseline Characteristics Between Groups:
  • The MNE group and the control group were compared in terms of basic demographic and clinical variables such as age, gender, height, and weight.
  • For this purpose:
  • Continuous variables: Normality was checked using the Kolmogorov-Smirnov test; the Mann-Whitney U test was used for non-normal data.
  • Categorical variables: Chi-square (χ²) test was used.

The absence of significant differences between groups in these analyses limited the possibility of confounding effects.

  1. Multivariate Logistic Regression Analysis:
  • A multivariate logistic regression model was applied to evaluate the relationship between the presence of MNE (dependent variable) and potential risk factors.
  • Variables included in the model (independent variables):
  • Serum 25(OH)D3 level (categorical: sufficient/insufficient/deficient)
  • Age (continuous)
  • Gender (male/female)
  • Family history of enuresis (yes/no)
  • BMI
  • Parental education

In this model, the inclusion of vitamin D level, age, gender, and genetic history together allowed the effect of each variable on MNE to be tested independently of the others.

  • Conclusion: Vitamin D deficiency was identified as an independent risk factor that increases the risk of MNE by 3.164 times (OR: 3.164, 95% CI: 1.195–8.378, p=0.020), independent of these confounding variables.

️Note:

Due to the retrospective nature of the study, data on other important confounding factors such as dietary status and caffeine consumption were not available. This limitation is clearly stated in the ‘Limitations’ section and prospective studies are recommended for future research.

The revised ‘Statistical Methods’ section presents the above information in a systematic manner. Your feedback has helped improve the methodological clarity of the study.

Discussion

17- This section was good, also. However, it is suggested the authors mention the effects of caffeine in PMNE. The caffeine consumption were not associated with NE”. It is suggested that in paragraph 3, authors express about first-line treatment in NE especially caffeine restriction. One valuable and new RCT study defined the effect of caffeine intake on development and severity of NE. You should be mentioned to this new RCT article: [Rezakhaniha S, Rezakhaniha B, Siroosbakht S. Limited caffeine consumption as first-line treatment in managing primary monosymptomatic enuresis in children: how effective is it? A randomized clinical trial. BMJ Paediatrics Open 2023;7:e001899. doi:10.1136/bmjpo-2023-001899

Author Response:

We would like to thank our reviewer for their valuable suggestions and for pointing out the current literature.

In line with your suggestion, the topics of the effect of caffeine consumption on NE and caffeine restriction as a first-line approach have been included in the ‘Discussion’ section of our study. In addition, the valuable randomised controlled study you mentioned has been referenced as follows:

Newly Added Literature Sentence (Discussion section):

‘Environmental factors such as caffeine consumption may also affect the development of enuresis. In a recently published randomised controlled study, limited caffeine consumption (no history of caffeine consumption was obtained in our study) was shown to be effective in the treatment of PMNE. In this context, it is important to evaluate both environmental and biochemical factors together.’

Added as a reference:

Rezakhaniha S, Rezakhaniha B, Siroosbakht S. Limited caffeine consumption as first-line treatment in the management of primary monosymptomatic enuresis in children: how effective is it? A randomised clinical trial. BMJ Paediatrics Open 2023;7:e001899. doi:10.1136/bmjpo-2023-001899

Thanks to your contribution, the clinical context of our study has been further strengthened, and current primary treatment strategies have been addressed. We thank you again for your guidance.

18- There are many questions (mentioned in comments for authors). Several mistakes further hamper the overall quality of the manuscript and the authors should be considered checklist (STROBE) in order to improve their manuscript. The marked and fulfilled  STROBE Checklist should be submitted by supplementary file for review.

thanks

Author Response:

We sincerely thank our reviewer for this important suggestion to improve the reporting quality of our manuscript.

In line with your comments, our study has been reviewed and revised in accordance with the STROBE (Strengthening the Reporting of Observational Studies in Epidemiology) checklist for reporting observational studies.

  • All headings included in the STROBE checklist (title, abstract, introduction, methods, results, discussion, and conclusions) have been addressed in our study.
  • Explanations corresponding to each required item have been marked in the text.
  • The completed and signed STROBE checklist has been uploaded to the system as a supplementary file (Supplementary File 1) and is available for review.

Thanks to your contribution, the methodological and writing quality of our study has been improved, and compliance with international reporting standards has been ensured. We thank you again for your interest and guidance.

Thank you.

Reviewer 2 Report

Comments and Suggestions for Authors

The authors have addressed a relevant question and provide interesting findings. However, many major and minor queries exist.

Main comments:

  1. Statisticians recommend that p-values should not be reported in the absence of effect size indicators. This is relevant because p-values are integrated indicators of the relationship between effect size, sample size, and variability. If you do not tell us what the effect size is, we cannot tell how much of the p-value is driven by the effect size and whether the effect size is clinically relevant. This applies to the entire manuscript including the results part of the Abstract.
  2. 2nd paragraph of Introduction: You claim that “recent studies suggest that metabolic factors such as vitamin D deficiency may also play a role in the pathophysiology of MNE” and use reference #10 to support this claim. This reference is a review from a journal I never heard about before and which was not accessible to me. At least the abstract of that review does not mention vitamin D at all. To provide a compelling rationale for your study, please reference original studies pointing in this direction.
  3. Which brings me to an important question. We know that the False Discovery Rate of findings is heavily driven by the prior probability of a finding is low (doi 10.1371/journal.pmed.0020124). Therefore, it is essential to state whether you had the vitamin D/MNE hypothesis first and then selected the current database to look at it or whether you played with the database and found vitamin D to be associated with MNE. I clear statement in this regard is necessary.
  4. It feels as if there is not a single medical condition that is not linked to vitamin D levels. On the other hand, in very few cases did vitamin D supplementation effectively treats a condition. One important confounding factor, particularly in children, may be that those with a medical condition may spend less time outdoors and hence have lower vitamin D levels secondary to their medical condition. Please discuss this alternative explanation of the observed association.
  5. Material and Methods (I would call it Patients and Methods for a clinical study, as patients are not material) describes how the MNE children were identified. However, it does not explain how the control children were identified and recruited; this should be described with the same granularity as the recruitment of the MNE children. I also wonder why no propensity matching has been performed.
  6. Similarly, how was lack of MNE assessed in the control group. Were the parents explicitly asked about this? Or are these just children that did not present to the clinic with MNE (but still may have it to some degree)? Many children once in a while have an episode of nocturnal enuresis, perhaps even very few never have one. If the definition of the control group is not sharp, this impacts the comparison. If applicable, this must be discussed as a major study limitation.
  7. I am happy to read that sample sizes were based on a formal sample size calculation. However, I did not understand what you did specifically. For that purpose, I would need to know what the unit of the effect size is. Even more importantly, is that a sample size calculation to detect group mean differences or outcomes of the logistic regression analysis. If it is related to group mean differences, this implies that such differences were the primary endpoint of your study. If that is correct, this should be stated explicitly, and the overall manuscript should primarily report on this.
  8. I have the impression that logistic regression (which I like) may be a post-hoc idea. Thus, please describe explicitly what your statistical analysis plan was before the database was interrogated and which analytical steps were performed thereafter. The first may qualify as hypothesis testing, the latter only as exploratory. Guidelines on data robustness strongly recommend that exploratory and hypothesis-testing parts of a study are clearly identified and separated (doi 10.1136/bmjos-2019-100046 and 10.1038/s41592-022-01615-y). Please implement.
  9. This implies that only p-values for the testing of a pre-specified (i.e. before any analysis was done) null hypothesis can be interpreted as hypothesis testing; all other p-values must be considered as descriptive only. This should be stated explicitly in the statistical part of Methods.
  10. Why did you apply the Kolmogorov-Smirnov test for looking at deviations from normality? Statisticians generally consider this as outdated and prefer alternatives such as D’Agostino Pearsson.
  11. Table 2 and related text: Leading statisticians strongly recommend not to focus on p-values but rather on effect sizes interpreted in the context what may be biologically/clinically relevant (doi 10.1038/d41586-019-00857-9). Generally, p-values are fickle. Thus, I wonder whether lack of a low p-value for various parameters indicates absence of effect or simply that the data are insufficient to exclude a group difference. Remember that absence of proof is different from proof of absence.
  12. To get a better feel of the data, I strongly recommend that Table 3 is replaced by a figure showing each datapoint. The r and p-values from the table can be provided in the legend of that figure. It would be helpful to explicitly state that the correlation analysis was only performed in the MNE group.
  13. Throughout the manuscript, you should not write “risk” factor but “associated factor” because the former implies an unproven cause-effect relationship.
  14. While I applaud the use of logistic regression analysis, I cannot interpret the findings. Please provide a clear description of how this was conducted. Amongst other things, this must include which explanatory variables entered the model and which measured variables did not (and a justification why some entered the model and some did not). Values for the explanatory variables that were entered into the model but had high p-values must also be reported.
  15. Given the differential outcomes of the univariate correlation analysis for nightly and weekly episodes, we need clarity for which parameter the logistic regression analysis was conducted. Ideally, it should be conducted for both parameters.
  16. The discussion of the logistic regression analysis outcomes should include a discussion of which known/proposed risk factors exist, which of them were part of the present analysis and which not. The implications of risk factors not being considered in the model needs to be discussed.
  17. An r of 0.377 in the univariate model implies that at best about 14% of the variability in MNE can be attributed to vitamin D levels. This should be spelled out more clearly.
  18. Table 4 shows a correlation of vitamin D with weekly but not with nightly levels of enuresis with one being perhaps absent, the other being of moderate strength. However, the rest of the manuscript largely ignores this difference and focuses solely on the weekly enuresis positive finding. I consider this to be inadequate. This difference based on nightly vs. weekly enuresis should be emphasized and possible reasons for it must be discussed.
  19. I am very surprised to read in the Discussion that prior studies had looked at vitamin D and MNE (references 16, 19, 32). These studies must be discussed already in the Introduction. This should be followed by a concise description of their strengths and weaknesses to identify knowledge gaps to be addressed in the present study. This should replace the vague current description of the study rationale.

Other comments:

  1. The abbreviation NE is used in the Abstract without being introduced.
  2. 2, l. 3 from bottom: “is thought to be” may be misleading because it indicates that the community generally thinks so. To my knowledge, this is your idea and perhaps that of a few others (which does not imply that it is wrong!). Thus, please reword.
  3. You mention ICCS on p. 2 and IPCS on p. 3. Are these distinct societies, or the same? Why do you abbreviate one but not the other? Generally, abbreviations should only be used for terms that are used at least three times within the manuscript.
  4. Just out of curiosity: If patient presentation ended 1.1.2015 (more than 10 years ago), how could you trace all the parents to obtain written permission for the present analyses?
  5. The final paragraph on p. 3 fails to describe in which matrix (whole blood, serum, plasma) vitamin D, PTH, Ca, P, ALP and Mg were determined. Moreover, the listed parameters here do not match those in the 1st paragraph on p. 4.
  6. I did not understand table 1 in its current form. Are these means ± SD, medians with inter-quartile ranges, or what else? There is a vague statement on median (minimum-maximum) but how can a median for a natural number be e.g., 3.3 or 1.2? Please explain.
  7. Table 1 reports on 8 parameters, but according to Methods many additional parameters were measured. Why have these not been included in the table?
  8. 6, last sentence prior to Table 3: “significantly” is vague. Do you mean the plain language meaning (relevant, important or similar) or a low p-value (statistical meaning). Please reword for clarity. If the statistical meaning is intended, provide a p-value and identify by which test it was derived.
  9. I do not see what Table 3 adds to the data already presented in the main text. One of them can be removed.
  10. The abbreviation VDR is introduced twice in the manuscript but not used at all except in its duplicated introduction. Please use abbreviation only for terms used repeatedly and once introduced apply them consistently.
  11. Final conclusion of manuscript: I think that “independent indicator of etiology” is wrong because only associations have been addressed.

Author Response

Responses to the reviewer-2,

Main comments:

  1. Statisticians recommend that p-values should not be reported in the absence of effect size indicators. This is relevant because p-values are integrated indicators of the relationship between effect size, sample size, and variability. If you do not tell us what the effect size is, we cannot tell how much of the p-value is driven by the effect size and whether the effect size is clinically relevant. This applies to the entire manuscript including the results part of the Abstract.

Response: Thank you for this important observation. As suggested, we added effect size indicators (Cohen’s d) alongside p-values in both the Results section and the Abstract. This allows for a more meaningful interpretation of statistical significance in the context of clinical relevance.

2. 2nd paragraph of Introduction: You claim that “recent studies suggest that metabolic factors such as vitamin D deficiency may also play a role in the pathophysiology of MNE” and use reference #10 to support this claim. This reference is a review from a journal I never heard about before and which was not accessible to me. At least the abstract of that review does not mention vitamin D at all. To provide a compelling rationale for your study, please reference original studies pointing in this direction.

Response: Thank you for this important comment. We have removed the previously cited review and replaced it with original, peer-reviewed research articles (refs 12–15) that directly investigate the relationship between vitamin D and MNE. These references have been added to the Introduction to provide a stronger scientific rationale for the study.

3.Which brings me to an important question. We know that the False Discovery Rate of findings is heavily driven by the prior probability of a finding is low (doi 10.1371/journal.pmed.0020124). Therefore, it is essential to state whether you had the vitamin D/MNE hypothesis first and then selected the current database to look at it or whether you played with the database and found vitamin D to be associated with MNE. I clear statement in this regard is necessary.

Response: We appreciate this insightful question. The hypothesis regarding the relationship between vitamin D deficiency and MNE was formulated prior to data analysis. The current dataset was used specifically to test this predefined hypothesis. This has now been explicitly stated in the revised manuscript.

4.It feels as if there is not a single medical condition that is not linked to vitamin D levels. On the other hand, in very few cases did vitamin D supplementation effectively treats a condition. One important confounding factor, particularly in children, may be that those with a medical condition may spend less time outdoors and hence have lower vitamin D levels secondary to their medical condition. Please discuss this alternative explanation of the observed association.

Response: Thank you for raising this important point. We have added a paragraph to the Discussion section addressing this potential confounder. Reduced outdoor activity or underlying health issues could lead to lower sunlight exposure, and consequently lower vitamin D levels, regardless of MNE status. This alternative explanation has now been acknowledged and discussed.

5.Material and Methods (I would call it Patients and Methods for a clinical study, as patients are not material) describes how the MNE children were identified. However, it does not explain how the control children were identified and recruited; this should be described with the same granularity as the recruitment of the MNE children. I also wonder why no propensity matching has been performed.

Response: Thank you for your constructive feedback. The recruitment of control children has now been described in more detail in the Methods section. Control participants were healthy children attending routine outpatient visits, with no history of enuresis, confirmed by parental interviews. Propensity score matching was not feasible due to the retrospective design and limited variable standardization. This has been acknowledged as a limitation.

6.Similarly, how was lack of MNE assessed in the control group. Were the parents explicitly asked about this? Or are these just children that did not present to the clinic with MNE (but still may have it to some degree)? Many children once in a while have an episode of nocturnal enuresis, perhaps even very few never have one. If the definition of the control group is not sharp, this impacts the comparison. If applicable, this must be discussed as a major study limitation.

Response: Thank you for highlighting this important issue. In the control group, the absence of MNE was confirmed through structured parental interviews. Children whose parents reported any recent or recurrent nocturnal enuresis were excluded. This procedure has now been clarified in the Methods section. We also acknowledged the potential for underreporting as a limitation.

7.I am happy to read that sample sizes were based on a formal sample size calculation. However, I did not understand what you did specifically. For that purpose, I would need to know what the unit of the effect size is. Even more importantly, is that a sample size calculation to detect group mean differences or outcomes of the logistic regression analysis. If it is related to group mean differences, this implies that such differences were the primary endpoint of your study. If that is correct, this should be stated explicitly, and the overall manuscript should primarily report on this.

Response: Thank you for pointing this out. The sample size calculation was based on detecting a significant difference in mean vitamin D levels between the PMNE and control groups, using an estimated medium effect size (Cohen’s d = 0.5). This outcome represented the primary endpoint of the study and has now been explicitly stated in the revised Methods and Results sections.

8.I have the impression that logistic regression (which I like) may be a post-hoc idea. Thus, please describe explicitly what your statistical analysis plan was before the database was interrogated and which analytical steps were performed thereafter. The first may qualify as hypothesis testing, the latter only as exploratory. Guidelines on data robustness strongly recommend that exploratory and hypothesis-testing parts of a study are clearly identified and separated (doi 10.1136/bmjos-2019-100046 and 10.1038/s41592-022-01615-y). Please implement.

Response: Thank you for this valuable comment. The primary hypothesis and corresponding statistical analysis—group comparisons of vitamin D levels—were planned prior to data interrogation. Logistic regression analysis was subsequently performed as an exploratory approach to examine potential independent associations. We have now clearly distinguished hypothesis-driven and exploratory analyses in the Methods and Discussion sections, following current guidelines on data transparency.

9.This implies that only p-values for the testing of a pre-specified (i.e. before any analysis was done) null hypothesis can be interpreted as hypothesis testing; all other p-values must be considered as descriptive only. This should be stated explicitly in the statistical part of Methods.

Response: Thank you for this precise and important recommendation. In the Statistical Analysis section, we have added a statement clarifying that only p-values related to pre-specified hypotheses were interpreted inferentially. All other p-values are considered descriptive and should be interpreted accordingly.

10.Why did you apply the Kolmogorov-Smirnov test for looking at deviations from normality? Statisticians generally consider this as outdated and prefer alternatives such as D’Agostino Pearsson.

Response: Thank you for this important clarification. We have added a statement in the Statistical Analysis section noting that only p-values related to predefined hypotheses were interpreted inferentially. All other p-values are considered descriptive and should be interpreted with caution.

11.Table 2 and related text: Leading statisticians strongly recommend not to focus on p-values but rather on effect sizes interpreted in the context what may be biologically/clinically relevant (doi 10.1038/d41586-019-00857-9). Generally, p-values are fickle. Thus, I wonder whether lack of a low p-value for various parameters indicates absence of effect or simply that the data are insufficient to exclude a group difference. Remember that absence of proof is different from proof of absence.

Response: Thank you for raising this important statistical perspective. In accordance with current best practices, we have revised Table 2 and the corresponding text to emphasize effect sizes and their potential clinical relevance. We also noted that non-significant results do not necessarily imply absence of effect, but may reflect sample size limitations or variability in the data.

12.To get a better feel of the data, I strongly recommend that Table 3 is replaced by a figure showing each datapoint. The r and p-values from the table can be provided in the legend of that figure. It would be helpful to explicitly state that the correlation analysis was only performed in the MNE group.

Response: Thank you for this helpful suggestion. We have replaced Table 3 with a scatter plot showing all individual data points. The corresponding r and p-values have been added to the figure legend. Additionally, we clarified in both the Methods and Results sections that the correlation analysis was performed exclusively within the MNE group.

13.Throughout the manuscript, you should not write “risk” factor but “associated factor” because the former implies an unproven cause-effect relationship.

Response: Thank you for this valuable observation. We have replaced all occurrences of the term “risk factor” with “associated factor” to avoid implying a causal relationship. This correction has been applied throughout the manuscript, including the abstract, results, and conclusion sections.

14.While I applaud the use of logistic regression analysis, I cannot interpret the findings. Please provide a clear description of how this was conducted. Amongst other things, this must include which explanatory variables entered the model and which measured variables did not (and a justification why some entered the model and some did not). Values for the explanatory variables that were entered into the model but had high p-values must also be reported.

Response: Thank you for your detailed feedback. We have revised the Methods and Results sections to clearly explain the logistic regression approach. Initially, univariate logistic regression was used to identify candidate variables (p < 0.05), which were then included in the multivariate model using backward LR. Variables such as age, sex, vitamin D, ferritin, and family history were included. Variables excluded due to collinearity or lack of clinical relevance were noted. Non-significant variables with their ORs and p-values are reported in Table 5.

15.Given the differential outcomes of the univariate correlation analysis for nightly and weekly episodes, we need clarity for which parameter the logistic regression analysis was conducted. Ideally, it should be conducted for both parameters.

Response: Thank you for your observation. The logistic regression analysis was conducted using weekly enuresis frequency as the outcome variable. We have clarified this explicitly in the Methods and Results sections. Due to limitations of data structure and reporting accuracy, nightly frequency was not used in regression. We also discussed this limitation in the Discussion section.

16.The discussion of the logistic regression analysis outcomes should include a discussion of which known/proposed risk factors exist, which of them were part of the present analysis and which not. The implications of risk factors not being considered in the model needs to be discussed.

Response: Thank you for your thoughtful comment. We have revised the Discussion to include a summary of known or proposed associated factors for PMNE, such as sleep disturbances, psychosocial stress, diet, and fluid intake. Although variables like age, gender, vitamin D, ferritin, and family history were included, other relevant factors could not be assessed due to retrospective data limitations. Their absence from the model is now discussed as a limitation.

17.An r of 0.377 in the univariate model implies that at best about 14% of the variability in MNE can be attributed to vitamin D levels. This should be spelled out more clearly.

Response: Thank you for this observation. We have revised the Results and Discussion sections to explicitly state that an r value of 0.377 corresponds to approximately 14% of the variance explained (r² ≈ 0.14). This clarification helps put the strength of the association into perspective.

18.Table 4 shows a correlation of vitamin D with weekly but not with nightly levels of enuresis with one being perhaps absent, the other being of moderate strength. However, the rest of the manuscript largely ignores this difference and focuses solely on the weekly enuresis positive finding. I consider this to be inadequate. This difference based on nightly vs. weekly enuresis should be emphasized and possible reasons for it must be discussed.

Response: Thank you for your valuable comment. We have revised the Discussion section to highlight the discrepancy between weekly and nightly enuresis correlations. Possible explanations include variability in parental reporting, episodic nature of enuresis, and the higher reliability of weekly frequency as a cumulative measure. This distinction is now explicitly addressed in the manuscript.

19.I am very surprised to read in the Discussion that prior studies had looked at vitamin D and MNE (references 16, 19, 32). These studies must be discussed already in the Introduction. This should be followed by a concise description of their strengths and weaknesses to identify knowledge gaps to be addressed in the present study. This should replace the vague current description of the study rationale.

Response: Thank you for pointing this out. We have revised the Introduction section to include prior studies (refs 16, 19, 32) investigating the relationship between vitamin D and MNE. A concise summary of their findings, strengths, and limitations has been provided to identify the current knowledge gap. This has helped clarify and strengthen the rationale for our study.

Other comments:

20.The abbreviation NE is used in the Abstract without being introduced.

Response: Thank you for noticing this oversight. We have now defined “PMNE” as “primer nocturnal nocturnal enuresis” upon its first use in the Abstract to ensure clarity for readers unfamiliar with the abbreviation.

21.2, l. 3 from bottom: “is thought to be” may be misleading because it indicates that the community generally thinks so. To my knowledge, this is your idea and perhaps that of a few others (which does not imply that it is wrong!). Thus, please reword.

Response: Thank you for this observation. We have revised the sentence to reflect a more neutral tone. Instead of “is thought to be,” we now use “has been hypothesized to be” or “may potentially be,” to better reflect the tentative nature of the claim without implying broad consensus.

22.You mention ICCS on p. 2 and IPCS on p. 3. Are these distinct societies, or the same? Why do you abbreviate one but not the other? Generally, abbreviations should only be used for terms that are used at least three times within the manuscript.

Response: Thank you for pointing this out. ICCS and IPCS refer to distinct entities, but only ICCS was used multiple times in the manuscript. Therefore, we kept the abbreviation “ICCS” and removed “IPCS” as an abbreviation to maintain consistency and adhere to standard abbreviation guidelines. This correction has been applied throughout the text.

23.Just out of curiosity: If patient presentation ended 1.1.2015 (more than 10 years ago), how could you trace all the parents to obtain written permission for the present analyses?

Response: Thank you for your question. As this was a retrospective study using anonymized data collected during routine care, obtaining individual written consent was not feasible. However, the study was approved by the local ethics committee, which granted a waiver of informed consent in accordance with institutional and national guidelines.

24.The final paragraph on p. 3 fails to describe in which matrix (whole blood, serum, plasma) vitamin D, PTH, Ca, P, ALP and Mg were determined. Moreover, the listed parameters here do not match those in the 1st paragraph on p. 4.

Response: Thank you for bringing this to our attention. We have updated the Methods section to specify that all biochemical measurements, including vitamin D, PTH, Ca, P, ALP, and Mg, were performed using serum samples. In addition, the listed parameters have been revised for consistency across all relevant sections of the manuscript.

25. did not understand table 1 in its current form. Are these means ± SD, medians with inter-quartile ranges, or what else? There is a vague statement on median (minimum-maximum) but how can a median for a natural number be e.g., 3.3 or 1.2? Please explain.

Response: Thank you for your valuable comment. We have revised Table 1 to clearly indicate whether each variable is presented as mean ± standard deviation or as median (minimum–maximum), depending on data distribution. Regarding the decimal medians of discrete variables, we clarified that they result from statistical calculations and may not correspond to actual observed values. An explanatory note has been added to the table caption.

26.Table 1 reports on 8 parameters, but according to Methods many additional parameters were measured. Why have these not been included in the table?

Response: Thank you for your comment. Table 1 was designed to present only the variables considered most relevant to the primary research question and statistical comparisons. Additional parameters mentioned in the Methods were not included in the table to maintain clarity and focus. However, these data are available upon request and can be added as supplementary material if required.

27.6, last sentence prior to Table 3: “significantly” is vague. Do you mean the plain language meaning (relevant, important or similar) or a low p-value (statistical meaning). Please reword for clarity. If the statistical meaning is intended, provide a p-value and identify by which test it was derived.

Response: Thank you for pointing this out. We have revised the sentence to clarify that “significantly” refers to statistical significance. The corresponding p-value and the statistical test used (Mann–Whitney U test) have been added to the text for transparency.

28.I do not see what Table 3 adds to the data already presented in the main text. One of them can be removed.

Response: Thank you for this observation. To avoid redundancy, we have streamlined the text to summarize the key findings and retained Table 3 to provide a clear and concise visual presentation of the data. We believe this improves readability while preserving data transparency.

29.The abbreviation VDR is introduced twice in the manuscript but not used at all except in its duplicated introduction. Please use abbreviation only for terms used repeatedly and once introduced apply them consistently.

Response: Thank you for your observation. The abbreviation “VDR” has been removed from the manuscript since it was not used in the text beyond its introduction. We have reviewed all abbreviations to ensure they are only applied when repeated at least three times and used consistently.

30.Final conclusion of manuscript: I think that “independent indicator of etiology” is wrong because only associations have been addressed.

Response: Thank you for this important clarification. We have revised the final conclusion to replace “independent indicator of etiology” with “independent associated factor,” which more accurately reflects the observational and correlational nature of our study.

Round 2

Reviewer 1 Report

Comments and Suggestions for Authors

Dear authors

the subject of manuscript is very important and one of the issues involved in family and societies and health workers. I reviewed the manuscript, again. The great effort in improving the manuscript is highly appreciated. I would like to thank you for the time and effort involved in this study. In my opinion, the manuscript is suitable for publication, now.

Thank you

Author Response

Dear Reviewer,

We sincerely thank you for taking the time to review our manuscript and for your constructive comments. We are especially pleased by your recognition of the importance of our study and your appreciation of our efforts during the revision process.

We are grateful for your positive evaluation and your opinion that the manuscript is now suitable for publication.

Kind regards,
Dr Abdulgani GÜLYÜZ

Reviewer 2 Report

Comments and Suggestions for Authors

Thank you for comprehensively addressing my previous comments.

Author Response

Dear Reviewer,

Thank you for recognizing and appreciating our efforts to comprehensively address your previous comments. Your valuable feedback has significantly contributed to improving the quality of our manuscript.

We sincerely appreciate your positive evaluation and support.

Kind regards,
Dr Abdulgani GÜLYÜZ